# Extra base hits: Widespread empirical support for instantaneous multiple-nucleotide changes

Alexander G. Lucaci[ID][�}, Sadie R. Wisotsky[�}, Stephen D. Shank[1], Steven Weaver[ID], Sergei L. Kosakovsky Pond*

Institute for Genomics and Evolutionary Medicine, Temple University, Philadelphia, PA, United States of America

[�} These authors contributed equally to this work.
* spond@temple.edu

**Data Availability Statement:** The study dataset is available on Zenodo, at https://doi.org/10.5281/zenodo.4570785.

## Abstract

Despite many attempts to introduce evolutionary models that permit substitutions to instantly alter more than one nucleotide in a codon, the prevailing wisdom remains that such changes are rare and generally negligible or are reflective of non-biological artifacts, such as alignment errors. Codon models continue to posit that only single nucleotide change have non-zero rates. Here, we develop and test a simple hierarchy of codon-substitution models with non-zero evolutionary rates for only one-nucleotide (1H), one- and two-nucleotide (2H), or any (3H) codon substitutions. Using over 42, 000 empirical alignments, we find widespread statistical support for multiple hits: 61% of alignments prefer models with 2H allowed, and 23%—with 3H allowed. Analyses of simulated data suggest that these results are not likely to be due to simple artifacts such as model misspecification or alignment errors. Further modeling reveals that synonymous codon island jumping among codons encoding serine, especially along short branches, contributes significantly to this 3H signal. While serine codons were prominently involved in multiple-hit substitutions, there were other common exchanges contributing to better model fit. It appears that a small subset of sites in most alignments have unusual evolutionary dynamics not well explained by existing model formalisms, and that commonly estimated quantities, such as dN/dS ratios may be biased by model misspecification. Our findings highlight the need for continued evaluation of assumptions underlying workhorse evolutionary models and subsequent evolutionary inference techniques. We provide a software implementation for evolutionary biologists to assess the potential impact of extra base hits in their data in the HyPhy package and in the Datamonkey.org server.

## Introduction

Most modern codon models in widespread use assume that any changes within a codon happen as a sequence of single instantaneous nucleotide changes, enforced by setting instantaneous rates between codons that differ in more than one nucleotides to zero. This choice was

**Funding:** This work was supported by National Institutes of Health grants R01 GM093939 and R01 AI134384 to SKP.

**Competing interests:** The authors have declared that no competing interests exist.

made independently for the mechanistic models of Muse and Gaut [1] and Goldman and Yang [2], and adopted by subsequent model developers and model users. For example, when Halpern and Bruno [3] introduced their mutation-selection models, they considered the general multi-hit (MH) case first, but then largely abandoned it, noting that the single hit reduction "..has very little effect on our results under the conditions we have investigated." This assumption is both computationally convenient and biologically sound in the majority of cases, since randomly occurring mutations "hitting" the same codon is a negligibly rare event. While these events are indeed rare, evidence for substitutions occurring in tandem at adjacent nucleotide sites had been reported at about the same time the codon models were being introduced [4].

Averof et al [5] reported significant rates of changes between TCN and AGY codon islands in perfectly conserved serine residues, and argued against going through intermediary non-synonymous changes due to their likely deleterious effects. Rogozin et al [6] took the opposite view, namely that strong purifying selection on single nucleotide changes is a more plausible explanation for such island hops in general.

Neither of those studies had considered an explicit evolutionary model, however. Serine is the only amino-acid with synonymous codon islands in the universal genetic code, but several other codes have other amino acids with this property: leucine in the *Chlorophycean* and *Scenedesmus obliquus* mitochondrial codes (TAG and CTH), and alanine in the *Pachysolen tannophilus* nuclear code (CTG and GCH).

Recent studies estimated that 2% of nucleotide substitutions are part of larger multiple nucleotide changes that occur simultaneously [7, 8], due in part to an error-prone DNA polymerase zeta. Human germline tandem mutations may constitute up to 0.4% of all mutations [9], and individual cases of such mutations have significant phenotypic consequences, e.g., via their effects on protein folding [10].

A number of codon model extensions have incorporated MH, invariably finding improvement in fit and (if the model allowed testing) statistically significant evidence of non-zero rates involving multiple nucleotide changes. Kosiol et al [11] developed a general MH empirical codon substitution model estimated jointly from a large collection of training alignments, and noted that it was overwhelmingly preferred to standard SH models on a sample of biological data from the PANDIT database.

Several groups have independently created alternative codon model parameterizations to allow for MH, including Whelan and Goldman [12] ("... these events [MH] are far more prevalent than previously thought"), Zaheri et al [13], and Dunn et al [14] (the latter two studies show a dramatically better model fit to empirical alignments when allowing MH). Other studies have used evolutionary models with varying degrees of accommodation for multiple hits [15–18]. Jones et al [19] implemented a complex model to detect adaptive evolution with a discrete-state phenotype, allowing for double and triple mutations to be absorbed into the parameter estimates. Despite multiple introductions to the field, these models have been unable to gain attraction in applied evolutionary analyses, and for some of these methods, software implementing them is no longer available.

Failure to include multiple hits in codon substitution models may mislead evolutionary hypothesis testing. Venkat et al [20] found that the addition of a double-hit rate parameter improved model fit and impacted branch-specific inferences of positive selection (MH along short branches can inflate false positives). Dunn et al [14] used principled simulation studies to show that fitting 1H models to data generated with low rates of multiple hits can increase false positive rates and dilute power for identifying individual sites subject to positive selection.

In this study, we develop simple extensions to the Muse-Gaut [1] codon model which add double, and triple instantaneous (2H, 3H, respectively) changes and compares them to simpler models in large collections of empirical data. Our models are mechanistic and simpler than those proposed by Whelan and Goldman [12] and Dunn et al [14]. This relative simplicity allows our models to be implemented and fitted quickly, and offers straightforward interpretation, including the ability to identify individual sites that benefit from the addition of MH. The primary goals of our data analyses is to establish how often evidence for multiple hits can be detected in large-scale empirical databases (something that no other study looking at evolutionary models has done), identify the codons that are frequently involved in such events, and explore plausible biological explanations for why these rates are non-zero for a majority of alignments.

## Materials and methods

### Substitution models

The most general model considered here is the 3H+ substitution model and all others can be derived from it as special cases (see Table 1 for key parameter notation). The model is a straightforward extension of the Muse-Gaut style of time-reversible, continuous Markov processes model [1]. In this study we compare five models:

1H.  is the standard Muse-Gaut style model which only permits single nucleotides to substitute instantaneously. Non-synonymous changes occur at rate $\omega$ (relative to the synonymous rate), and this rate varies from site to site according to a three-bin general discrete distribution (GDD).

2H.  is the 1H model extended to allow two nucleotides in a codon to substitute instantaneously with rate $\delta$ (relative to 1H synonymous rate).

3HSI.  is the 2H model extended to allow three nucleotides in a codon to substitute instantaneously if the change is synonymous (e.g., serine islands), with relative rate $\psi_s$.

3H.  is the 2H model extended to also permit any three-nucleotide substitutions, with relative rate $\psi$.

3H+.  is the 3HSI model extended to also permit any three-nucleotide substitutions, with relative rate $\psi$.

All codon substitutions in these models fall into one of six categories defined by (i) whether or not they are synonymous or non-synonymous, and (ii) by how many nucleotides are being replaced (1,2,3). The instantaneous rate expression for substitutions between codons $i$ and $j$ ($i \neq j$) for these six classes, and how many of all possible codon substitutions are in each class,

**Table 1. Key parameters of our models.** Estimation strategies for each parameter for the five different models are shown.

| Parameter | Description | Model | | | | |
|---|---|---|---|---|---|---|
| | | 1H | 2H | 3HSI | 3H | 3H+ |
| $\omega_i$ | Site-specific dN/dS ratio | Random effect 3-bin GDD distribution | | | | |
| $\delta$ | Global 2H/1H rate ratio | 0 | Estimated | Estimated | Estimated | Estimated |
| $\psi_s$ | Global 3H/1H rate ratio for synonymous codon islands | 0 | Estimated | Estimated | $= \psi$ | Estimated |
| $\psi$ | Global 3H/1H rate ratio | 0 | 0 | 0 | Estimated | Estimated |

Abbreviations: GDD = general discrete distribution; 1H, 2H, 3H = instantaneous changes involving one, two, or three nucleotides.

**Table 2. Expressions for different types of susbtitutions in the model rate matrix.**

| Type | Expression for $q_{ij}$ | Example | Count | |
|---|---|---|---|---|
| | | | Universal | mtDNA |
| 1H synonymous | $\theta_{ij}\pi_j$ | $ACA \rightarrow ACT: \theta_{CT}\pi_T^3$ | 134 | 128 |
| 1H non-synonymous | $\omega^k\theta_{ij}\,\pi_j$ | $AAA \rightarrow AGA: \omega^k\theta_{AG}\pi_G^2$ | 392 | 380 |
| 2H synonymous | $\delta\prod\limits_{n=1}^{2}\theta_{ij}^n\pi_j^n$ | $CTC \rightarrow TTA: \delta\theta_{CT}\theta_{AC}\pi_T^1\pi_A^3$ | 28 | 16 |
| 2H non-synonymous | $\delta\omega^k\prod\limits_{n=1}^{2}\theta_{ij}^n\pi_j^n$ | $AAA \rightarrow ACC: \delta\omega^k\theta_{AC}\theta_{AC}\pi_C^2\pi_C^3$ | 1540 | 1500 |
| 3H synonymous | $\psi_s\prod\limits_{n=1}^{3}\theta_{ij}^n\pi_j^n$ | $AGC \rightarrow TCA: \psi_s\theta_{AT}\theta_{CG}\theta_{AC}\pi_T^1\pi_C^2\pi_A^3$ | 12 | 12 |
| 3H non-synonymous | $\psi\omega^k\prod\limits_{n=1}^{3}\theta_{ij}^n\pi_j^n$ | $GTG \rightarrow TAC: \psi\omega^k\theta_{GT}\theta_{AT}\theta_{CG}\pi_T^1\pi_A^2\pi_C^3$ | 1554 | 1504 |

Six cases for instantaneous rates $q_{ij}$ of substituting codon $i$ with codon $j$ ($i \neq j$). The "Count" columns shows the number of rate matrix entires in each class (excluding the diagonal) for two commonly used genetic codes.

are shown in Table 2. In addition to key model parameters defined in Table 1, the model contains a number of other, standard parameters, which are not the main focus of inference and can be viewed, for the most part, as nuisance parameters. They include $\theta_{ij}$: nucleotide-level biases coming from the general time reversible model (5 parameters), and $\pi_j$ are codon-position specific nucleotide frequencies estimated from counts using the CF3x4 procedure [21]. $\omega^k$ are non-synonymous / synonymous rate ratios which vary from site to site using a random effect ($D$-bin general discrete distribution, $D = 3$ by default, $2D - 1$ parameters). The key parameters are global relative rates of multiple hit substitutions: $\delta$ is the rate for 2H substitutions relative to the synonymous 1H rate (baseline), $\psi$—the relative rate for non-synonymous 3H substitutions, and $\psi_s$—the relative rate for synonymous 3H substitutions. All parameters, except $\pi$, including branch lengths are fitted using directly optimized phylogenetic likelihood in HyPhy [22]. Initial estimates for branch lengths and $\theta$ are obtained using the standard nucleotide general time reversible model. Following this initialization, models are fitted in the order of increasing complexity (1H, then 2H, then 3HSI, then 3H+), using parameter estimates from from each stage as initial points for the next stage.

## Site-level support for MH

In order to identify which individual sites show preference for MH models, we use evidence ratios (ER), defined as the ratio of site likelihoods under two models being compared, e.g., $ER(2H : 1H) \coloneqq \frac{L(s_i|2H)}{L(s_i|1H)}$. We previously showed that ER are useful for identifying the sites driving support for one model over another [23], and they incur trivial additional overhead to compute once model fits have been performed.

## Hypothesis testing

Nested models are compared using likelihood ratio tests with $\chi_d^2$ asymptotic distribution used to assess significance. The degrees of freedom ($d$) parameter is as follows: $d = 1$ for 2H:1H, 3SHI:2H, and 3H+:3HSI comparisons; $d = 2$ for 3H+:2H comparison; $d = 3$ for 3H+:1H comparison.

## How do our models relate to previously published multiple-hit models?

The BS+MNM model [20] was designed for testing subsets of branches for episodic diversifying selection. It is very similar to our 2H model, except that $\theta_{ij}$ in their model follows the HKY85 parameterization, it is possible to allow $\kappa$ (transition/transversion ratio) to be different between 1H and 2H changes, and target codon frequencies are used in $q_{ij}$ [2]. The Empirical Codon Model or ECM [11] directly estimates numerical rates for all pairs of codon exchanges in the GY94 frequency framework from a large training dataset. The SDT model [12] uses a context-averaging approach to include the effect of substitutions that span codon boundaries, and is difficult to directly relate to our models; the 3H model might be the closest to the SDT model. Regrettably, there does not seem to exist a working implementation of the SDT model (pers. comm from Simon Whelan), which makes direct comparison to our approaches impractical. The KCM model [13] only has a single rate for multiple hits (double or triple), and has position-specific nucleotide substitution rates ($\theta$ in our notation), so it would be most comparable to the 3H model with $\delta = \psi$. The GPP model class [14] can be parametrized to recapitulate our models because it can capture (in a log-linear parametric form) arbitrary rate matrices with suitable parametric complexity. Several of the models in the GPP class include multiple hits, but they are not directly comparable to ours, mostly because they also incorporate $\omega$ rates that depend on physicochemical properties of amino acids, and because the exact parametric form of the models are hard to glean from available description.

## Empirical data

The Moretti et al, 2014 (Selectome) data collection consists of 13,714 gene alignments from the *Euteleostomi* clade of Bony Vertebrates from Version 6 of the database [24] and can be downloaded from data.hyphy.org/web/busteds/. The Shultz et al data collection [25] contains 11,262 orthologous protein coding genes from 39 different species of birds and is freely available at https://datadryad.org/stash/dataset/doi:10.5061/dryad.kt24554. The Enard et al data collection [26] includes 9,861 orthologous coding sequence alignments of 24 mammalian species and is available at https://datadryad.org/stash/dataset/doi:10.5061/dryad.fs756. Our mtDNA dataset consists of both invertebrate and vertebrate Metazoan orders with mitochondrial gene alignments. This dataset was originally published in Mannino et al [27], and can be found at https://github.com/srwis/variancebound.

## Simulated data

We generated simulated alignments of two sequences in HyPhy using the `SimulateMG94` package from https://github.com/veg/hyphy-analyses/. These alignments were simulated under the 1H (no site-to-site rate variation) with varying sequence and branch lengths as well as varied but constant $\omega$ across sites but no multiple hits. We created 1000 simulations scenarios to capture a range of important model parameters and drew 5 replicates per scenario. $\omega$ was drawn uniformly from $U(0.01, 2.0)$, branch length was drawn $Exp(U(0.01, 1.0))$, and codon lengths as an integer from 100 to 50000 uniformly. Parameter values were sampled using the Latin Hypercube approach to improve parameter space coverage.

Multiple sequence simulations were based on the fits to one of four benchmark datasets: Drosophila *adh*, Hepatitis D antigen, HIV *vif*, and the Vertebrate rhodopsin data. We took all model parameters estimated under the 3H+ model as the starting point, and generated 500 replicates per dataset of which 35% were null (1H), 10% each from 2H, 3SHI or restricted 3H+ ($\psi_s = 0$), and 35% from 3H+. $\delta$, $\psi$ and $\psi_s$ parameters, when allowed to be non-zero by the model, were sampled from $U(0, 1)$, $U(0, 1)$, and $U(0, 10)$, respectively.

**Table 3. Analysis of 13 benchmark datasets for evidence of multi-nucleotide substitutions.**

| Gene | N | S | T | $\delta$ | $\psi_s$ | $\psi$ | LRT p-value | | | | | # sites with $ER > 5$ | |
|---|---|---|---|---|---|---|---|---|---|---|---|---|---|
| | | | | | | | 2H:1H | 3H+:1H | 3H+:2H | 3H+:3HSI | 3HSI:2H | 2H:1H | 3H+:2H |
| $\beta$-globin | 17 | 144 | 2.5 | 0.7 (0.81) | >100 | 0 | **<0.001** | **<0.001** | **<0.001** | 1 | **<0.001** | 10 | 6 |
| Flavivirus NS5 | 18 | 342 | 6 | 0.49 (0.73) | 2.3 | 0.6 | **<0.001** | **<0.001** | 0.056 | 0.062 | 0.13 | 16 | 0 |
| Primate Lysozyme | 19 | 130 | 0.24 | 0 (0) | 0 | 0 | 1 | 1 | 1 | 1 | 1 | 0 | 0 |
| COXI | 21 | 510 | 5.3 | 0.4 (0.4) | 0 | 0 | **<0.001** | **0.0018** | 1 | 0.94 | 0.98 | 3 | 0 |
| Drosophila *adh* | 23 | 254 | 1.4 | 0.31 (0.4) | 0 | 0.42 | **<0.001** | **<0.001** | 0.19 | 0.067 | 0.99 | 4 | 0 |
| Encephalitis *env* | 23 | 500 | 0.84 | 0.076 (0.076) | 0 | 0 | 0.19 | 0.42 | 1 | 1 | 0.98 | 0 | 0 |
| Sperm lysin | 25 | 134 | 2.8 | 0.4 (0.46) | 2.3 | 0.3 | **<0.001** | **<0.001** | **0.04** | **0.015** | 0.49 | 21 | 1 |
| HIV-1 *vif* | 29 | 192 | 0.96 | 0.007 (0.044) | 0 | 0.17 | 0.058 | **0.0013** | **0.0077** | **0.0018** | 0.95 | 0 | 2 |
| Hepatitis D virus antigen | 33 | 196 | 1.9 | 0.34 (0.37) | 0 | 0.2 | **<0.001** | **<0.001** | 0.25 | 0.098 | 0.99 | 15 | 0 |
| Vertebrate Rhodopsin | 38 | 330 | 3.9 | 0.54 (0.72) | 9.2 | 0.9 | **<0.001** | **<0.001** | **<0.001** | **<0.001** | **0.0029** | 43 | 3 |
| Camelid VHH | 212 | 96 | 15 | 0.29 (0.32) | 0 | 0.13 | **<0.001** | **<0.001** | **0.011** | **0.0026** | 0.92 | 46 | 0 |
| Influenza A virus HA | 349 | 329 | 1.4 | 0.06 (0.06) | 0 | 0.0093 | **<0.001** | **<0.001** | 0.95 | 0.74 | 0.98 | 5 | 0 |
| HIV-1 *RT* | 476 | 335 | 6.6 | 0.086 (0.093) | 0 | 0.048 | **<0.001** | **<0.001** | 0.15 | 0.052 | 1 | 17 | 1 |

*N*—number of sequences, *S*—number of codons, *T*—total tree length (expected subs/site) under the 1H model, two-hit ($\delta$) rate estimate under the 3H model (2H model in parentheses), there-hit synonymous island date ($\psi_s$) estimate under the 3H model, three-hit rate ($\psi$) estimate under the 3H model. Likelihood ratio test p-values for all pairs of nested models e.g. 2H:1H—2H alternative, 1H null. Values <0.05 are bolded. # sites with $ER > 5$ lists the number of sites which show strong preferences for 2H or 3H model using evidence ratios of at least 5 (see text).

Sequences with indel rate variation were generated using INDELible v1.03 [28]. We varied indel rates between 0.01 to 0.06 in increments of 0.005 (100 replicates per value), and the modeled site-to-site rate variation a 3-bin M3 model.

## Implementation

All analyses were performed in HyPhy version 2.5.1 or later [22]. The `fmm` (FitMultihitModel) module used to fit the standard 1H model along with 2H, 3H and 3HSI versions is available from: https://github.com/veg/hyphy-analyses/, and is a part of the standard library (invoked with `hyphy fmm`) in HyPhy version 2.5.7 or later and also in our `datamonkey.org` server [29]. The result of an `fmm` analysis is a JSON file which be visualized using a web-application at http://vision.hyphy.org/multihit.

## Results

### Benchmark alignments

We introduce the models using a collection of thirteen representative alignments Table 3 that we and others have recently used to benchmark selection analyses [30]. We also include a primate lysozyme alignment originally analyzed with early codon models by Yang [31]. We consider five models (see Table 1 and the methods section for details), which form a nested hierarchy (with the exception of 3HSI and 3H which are not nested), each with one additional alignment-wide parameter.

1. **Evidence for multiple hits is pervasive.** In ten of thirteen datasets the analyses strongly reject the hypothesis that 2H have zero rates, with $p < 0.001$ (2H:1H comparison). For five of thirteen datasets, we can further reject the hypothesis that 3H have zero rates (3H+:2H comparison) at $p \leq 0.05$.

2. **Varied patterns for rate preferences.** Even in this small collection of datasets, the entire spectrum of options is present. For the Primate Lysozyme dataset there is no evidence for anything other than 1H changes, while for the Vertebrate Rhodopsin dataset each of the individual rates is significantly different from 0. HIV-1 *vif* dataset is the only dataset that does not support 2H rates, but does support 3H rates. Five datasets share a pattern: reject 1H in favor of 2H, and 1H in favor of 3H+, but none of the others, which can be interpreted as support for 2H rates, but none of the 3H rates.

3. **Varied extent of site-level support for MH.** Ratios between site-level likelihoods under individual models, denoted here as ER (evidence ratios), can indicate which model provides better fit to the data at a particular site. The number of sites with strong ($ER > 5$) preference for 2H vs 1H model was positive for all models rejecting 1H in favor of 2H with LRT, and ranged from 3 to 46, while a smaller number of sites ($0 - 6$) preferred 3H+ to 1H. Interestingly, for Camelid VHH, where the LRT rejects 1H in favor of 3H+, no individual sites had $ER > 5$, implying that the support for this model came from a number of individual weak site contributions.

4. **Interaction between 1H, 2H and 3H rates.** Assuming that the biological process of evolution does include MH events, not modeling them appropriately might have the effect of *inflating* other rate estimates.
   In line with other studies [14], the addition of 2H rates lowers the point estimates of $\omega$ rates for all datasets where 2H:1H comparison is significant at $p \leq 0.05$ (S1 Table), sometimes dramatically (e.g., by a factor of 0.6× for the $\beta$−globin gene). This could be indicative of estimation bias due to model misspecification. Similarly, the $\delta$ rate under the 2H model is always higher than the rate estimate under the 3H+ model, implying that the 2H rate may be "absorbing" some of the 3H variation. We will later see the same pattern emerge in large-scale sequence screens.

To bolster our intuitive understanding of model preferences, we visualized inferred substitutions at four archetypal sites in the Vertebrate Rhodopsin alignment [32], for which every single rate in the 3H+ model was significantly non-zero (Fig 1). We used joint maximum likelihood ancestral state reconstruction [33] under the 3H+ model to estimate the number and kind of substitutions that occurred at each site (this number is a lower bound and is subject to estimation uncertainty; here we use it for illustration purposes).

Single-hit site. Site 37 is what one might call a traditional single-hit substitution site, where the 1H model is preferred to all other models based on ER values; all apparent substitutions involve changes at a single nucleotide, hence the standard 1H is perfectly adequate. Of 330 codons, 149 had a preference for the 1H model compared to the 2H model.

Two-hit site. Site 144 has a dramatic preference for the 2H model over the 1H model ($ER > 300$); of 6 total substitutions, 4 involved a change at 2 nucleotides (and none—at 3).

Serine island site. Site 281 has a preference for the 3HSI model over the 2H model ($ER = 39$), and has a complex substitution pattern: nine 1H, four 2H, and two 3H substitutions; both 3H substitutions at this site involve synonymous changes between serine codon islands (TCN and AGY). 148 other sites had a preference ($ER > 1$) for 3HSI over 2H.

Three-hit site. Site 236 prefers 3H to 3HSI ($ER = 5.4$) as the only apparent 3H substitution at that site does not involve serine.

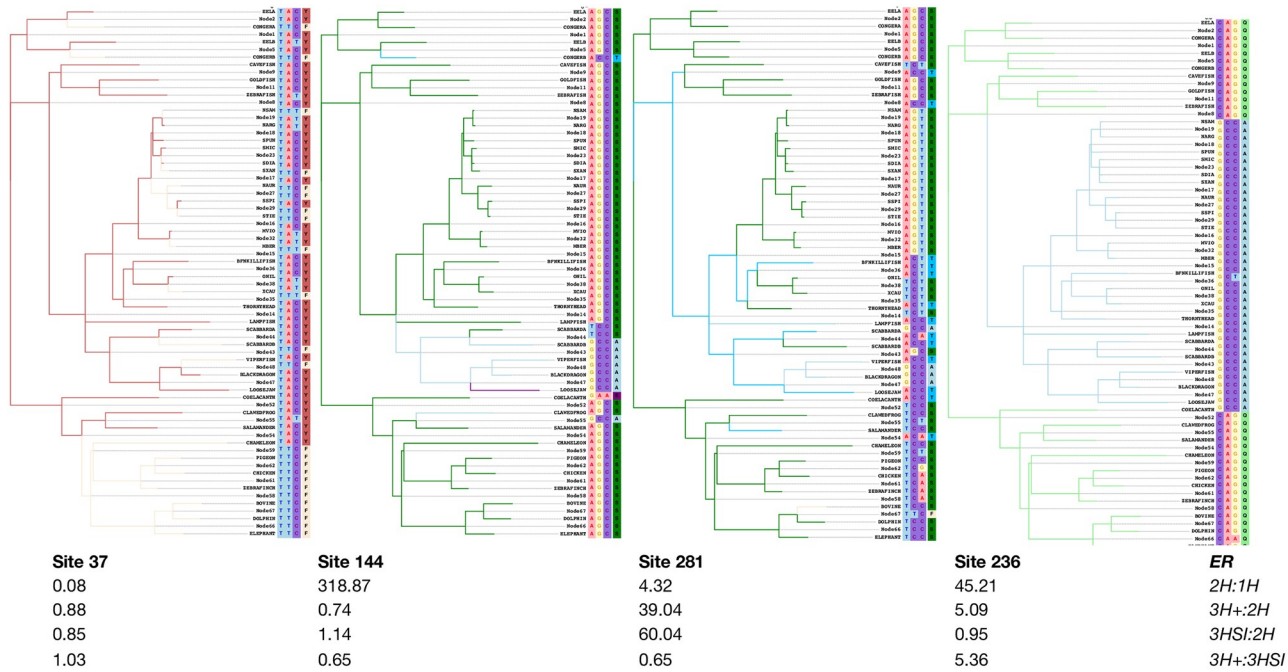

**Fig 1. Archetypal sites based on model preferences.** Four alignment sites from the Vertebrate Rhodopsin dataset [32] chosen to illustrate substitution patterns which give rise to support for specific rate models. Branches are colored by the amino-acid that is observed/estimated to exist at the end of the branch. Internal nodes are labeled with ancestral states inferred under the 3H+ model. Evidence ratios, which are the ratios of MLE site likelihoods under the respective models, for four pairwise model comparisons are listed below each site.

## Large-scale empirical analyses

We fitted the hierarchy of MH models to 42, 498 empirical datasets, assembled from three large-scale studies of natural selection of nuclear genes [24–26], and a smaller collection vertebrate and invertebrate mitochondrial genes [27], which represent a different evolutionary landscape (e.g., not affected by polymerase zeta).

**Strong evidence for non-zero multiple-hit rates.** We found widespread statistical support for models that include non-zero rates involving multiple nucleotides. The 1H model was overwhelmingly rejected in favor of the 2H model (Table 4), and the improvement in fit was quite dramatic on average, for all but the Enard et al [26] collection. A substantial fraction of

**Table 4. Evidence for multiple hit rates in empirical datasets.**

| Dataset | 2H:1H | 3H+:2H | 3H:1H | 3H+:3HSI | 3HSI:2H |
|---|---|---|---|---|---|
| Invertebrate mtDNA | 92% (119.2) | 7.4% (17.12) | 92% (122.2) | 8.9% (19.97) | 2.3% (9.089) |
| Vertebrate mtDNA | 54% (33.30) | 3.0% (16.60) | 50% (36.92) | 3.2% (15.11) | 0.69% (7.986) |
| Shultz *et al* (2009) | 62% (32.39) | 20% (17.76) | 63% (39.87) | 21% (13.63) | 7.4% (12.84) |
| Moretti *et al* (2014) Unmasked | 76% (55.99) | 37% (21.82) | 77% (67.67) | 20% (13.56) | 29% (16.73) |
| Moretti *et al* (2014) Masked | 76% (67.35) | 32% (27.3) | 74% (83.23) | 17% (18.24) | 23% (20.92) |
| Enard *et al* (2006) | 28% (15.69) | 5.4% (14.18) | 28% (20.39) | 5.3% (10.49) | 3.4% (11.07) |
| Overall | 61.02% (53.99) | 23.07% (19.13) | 60.79% (61.71) | 15.66% (15.17) | 15.01% (13.11) |

For each collection of alignments, the table shows the fraction with significant ($p < 0.01$, based on a 5-way conservative Bonferroni correction for FWER of 5%) LRT test results, and the average value of the likelihood ratio test statistic (for significant tests) in parentheses. Masked vs unmasked refers to the two versions of data in Moretti *et al* (2014): alignments in the masked version have some low quality sites removed.

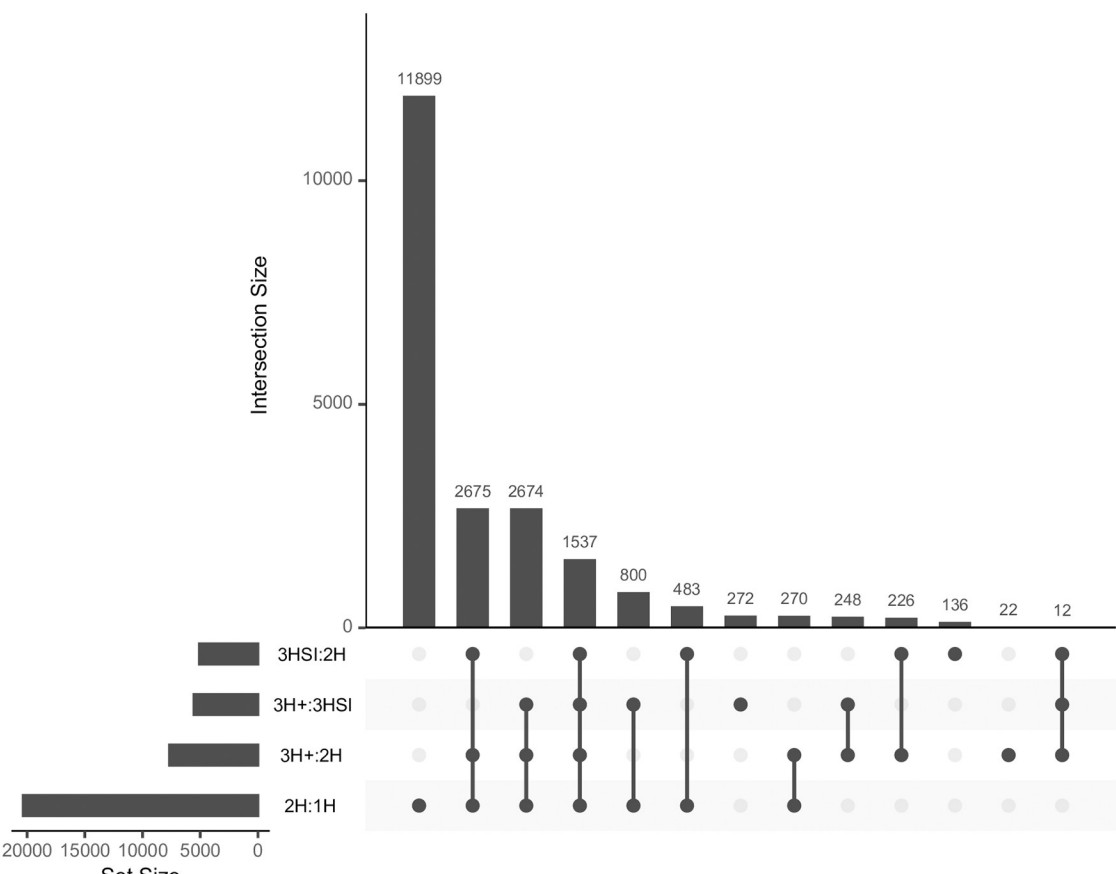

**Fig 2. Intersections of likelihood ratio test significance.** Overlaps of empirical alignments with $p \leq 0.01$ according to each of four LRTs performed for the combined empirical datasets. Groups of alignments for which a particular combination of tests was significant are shown in the main chart. The auxiliary chart in the lower left shows the number alignments belonging to a particular comparison category (row "sums"), with the significant tests indicated with filled dots. For example, there are 1537 alignments where all 4 tests are significant, and 136 alignments where the only significant test is 3SHI:2H.

alignments preferred models that allowed non-zero three-hit rates over the two-hit model, and also the 3H+ model which does not limit 3H instantaneous changes to only synonymous codons. Based on the results of the four likelihood ratio tests, each dataset could be assigned to a unique *rate preference* category Fig 2. For example, 11, 899 alignments preferred 2H to 1H model, but none of the other comparisons were significant, i.e there was no evidence for non-zero 3H instantaneous rates. 2, 675 alignments preferred 2H to 1H, and 3H+ to 2H, i.e. provided evidence for non-zero 3H instantaneous rates. 483 alignments preferred 2H to 1H and 3HSI to 2H, but not 3H+ to 3HSI, implying that all 3H changes were constrained to synonymous codon islands.

**Factors associated with MH detection.** The rates at which 2H, 3H and 3HSI rates were detected with $p < 0.01$ as functions of simple statistics of the alignments, are shown in Fig 3. Larger (more sequences) and longer (more codons) alignments generally elicited higher detection rates for all types of multiple hits. Increasing overall divergence levels between sequences, measured by the total tree length, also corresponded to increasing detection rates, up to a saturation point. The mean strength of selection, measured by the gene-average $\omega$ had little effect on detection rates, except for a dip in the tail. In a simple logistic regression using 2H:1H $p < 0.01$ as the outcome variable, sequence length, and number of sequences were positively

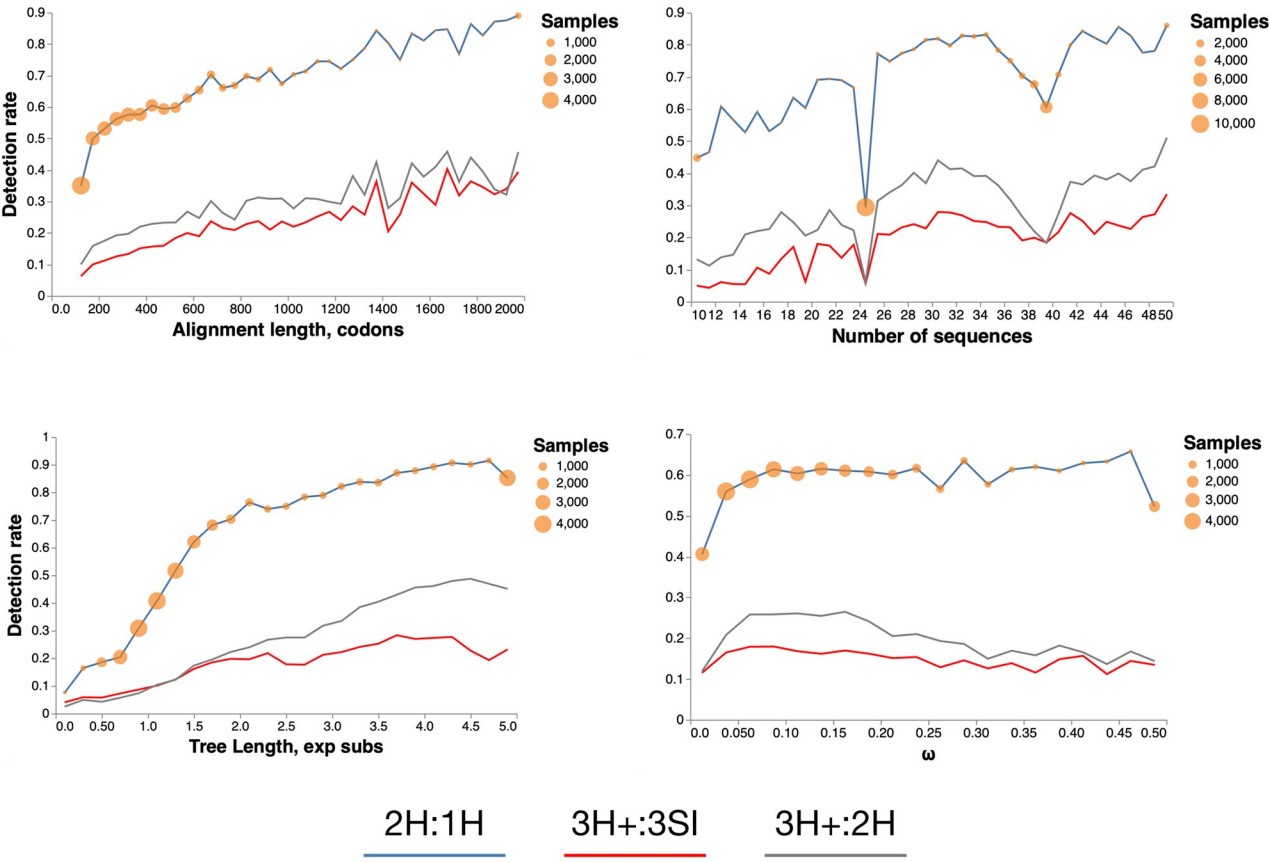

**Fig 3. Multiple hit detection rate.** The fraction of alignments where the corresponding test was significant at $p \leq 0.01$ as a function of one of four alignment properties. Orange circles depict the binning steps and the number of alignments falling into each bin. For tree lengths and $\omega$ values we used estimates under the 1H model.

associated with the detection rate ($p < 0.0001$), while tree length was confounded with the number of sequences and was not independently predictive, and $\omega$ was not significantly predictive.

**Strong MH signal comes from a small fraction of sites.** For alignments where there was significant evidence for non-zero 2H and/or 3H rates ($p < 0.01$), a small fraction of sites strongly ($ER > 5$) supported the corresponding MH model. For the 2H:1H comparison, a median of 0.67% (interquartile range, IQR [$0.21\% - 1.7\%$]), and for the 3H:2H comparison, a median of 0.52% (IQR [$0.26\% - 0.94\%$]) (S2 Fig) sites in an alignment had high ER in support of the respective model.

**Patterns of substitution associated with MH rates.** Substitutions between serine islands (AGY and TCN) appear to be the most frequent inferred 3H change in biological alignments (see Fig 4). Six of the most common substitutions at sites with high ER in support of the 3H + model involve island jumping. However, other amino-acid pairs are also involved in hundreds of apparent substitutions, e.g. $ATG(M) \leftrightarrow GCA(A)$. Of the 7664 datasets that reject the 2H model in favor of the general 3H+ model, 2901(37.9%) fail to reject 3HSI in favor of 3H+, implying that they only require non-zero rates for synonymous island jumps. However, many of the same changes frequently appear at sites that do not strongly prefer 3H+ to 2H model, but strongly prefer 2H to 1H model (i.e, 2H sites). A key determinant of whether or not an AGY:TCN or other 3H change benefits from non-zero $\psi$ rates is the length of the branch

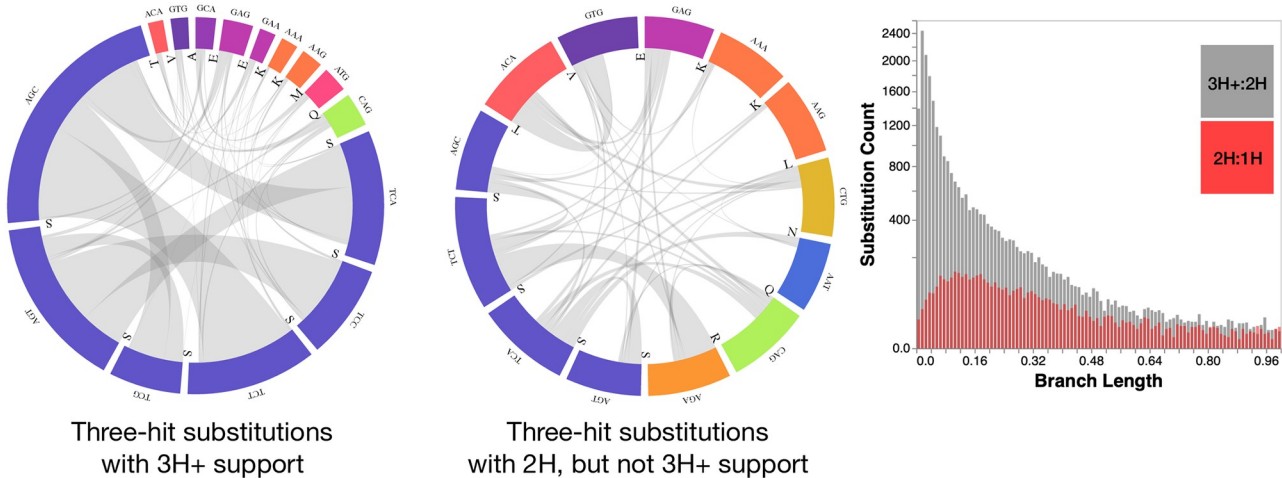

**Fig 4. Three-hit substitutions commonly occurring in empirical data.** A subset of common three-hit substitutions across all empirical datasets. Three-hit substitutions with 3H+ support are defined as those occurring at sites with $ER(3H+:2H) > 5$. Three-hit substitutions with 2H but not 3H + support are defined as those occurring at sites with $ER(3H+:2H) < 1$ and $ER(2H:1H) > 5$. Branch lengths along which the two types of substitutions are inferred to occur are shown in the histogram.

where the change is inferred to occur. Branches with 3H changes that supported 3H+ model were significantly shorter than those where 2H model was sufficient: median 0.09 substitutions/site, vs median 0.26 substitutions/site. Consequently, the need to explain 3H changes happening over short branches (shorter evolutionary time, slower overall rates) provides evidence in support of 3H+ models.

Among 3H non-synonymous substitutions (see Fig 5) codons encoding for serine are still prominently represented, but not as dominant, with numerous substitutions involving methionine and other amino-acids.

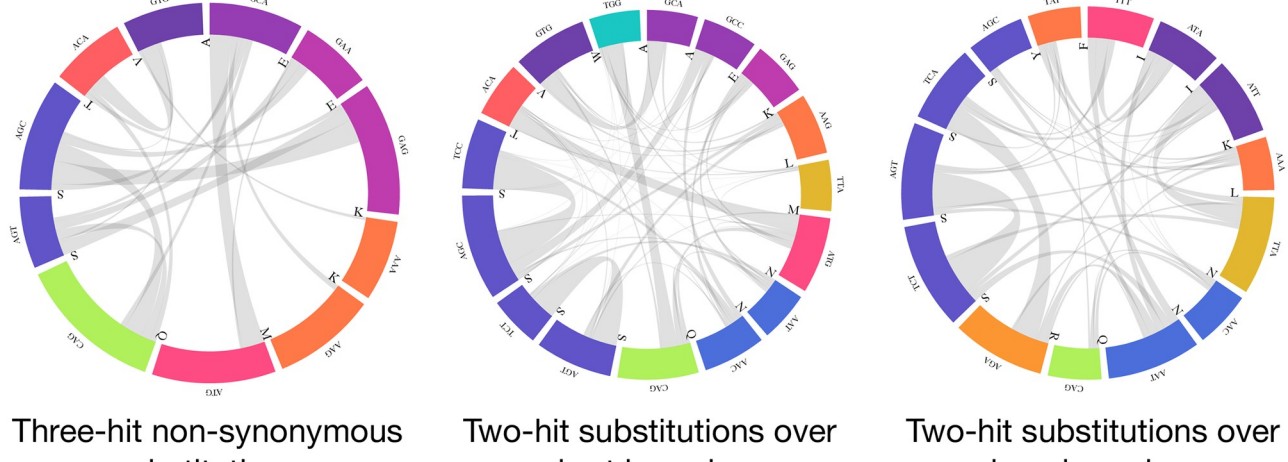

**Fig 5. Three-hit non-synonymous substitutions and two-hit substitutions occurring in empirical data.** A subset of common substitutions across all empirical datasets. Three-hit substitutions with non-synonymous support are defined as those occurring at sites with $ER(3H+:3HSI) > 5$. Two-hit substitutions over short branches are defined as those occurring at sites with $ER(3H+:2H) < 1$ and $ER(2H:1H) > 10$ and branch length is $\leq 0.05$ subs/site. Two-hit substitutions over short branches are defined as those occurring at sites with $ER(3H+:2H) < 1$ and $ER(2H:1H) > 10$ and branch length is $\geq 0.25$ subs/site.

Serine codons are similarly frequently involved in 2H substitutions, along both short and long branches (e.g. between codons such as $AGC \leftrightarrow TCC$ and $AGT \leftrightarrow TCT$), but other pairs are exchanged at least 90 times, including $ACA(T) \leftrightarrow ATG(M)$ and $CAG(Q) \leftrightarrow TGG(W)$ (short branches) and $ATT(I) \leftrightarrow TTA(L)$ (long branches).

**Interaction between rate estimates.** As with the benchmark datasets, the inclusion of multiple hit rates in models has an effect on other substitution rates. The gene wide point-estimate of $\omega$ is systematically lowered by the inclusion of non-zero $\delta$ rates, even though there are rare instances when the $\omega$ estimates are increased (S1 Fig). A Theil Sen robust linear regression estimate yields $\omega(2H) \sim 0.965 \times \omega(1H)$, but for 1150(5.7%) of the datasets with where 2H:1H comparison was signifiant, the $\omega(2H) < 0.75 \times \omega(1H)$. Consequently the estimation bias in important evolutionary rates due to model misspecification for some of the datasets could be significant. The inclusion of 3H components in the model lowers the 2H rate as well $\delta(3H+) \sim 0.77 \times \delta(2H)$.

**Impact on branch length estimates.** Branch lengths estimated with 2H and 3H models on the Selectome dataset were, on average, 0.93× the length of the standard (1H) estimate, while they were effectively identical between 2H and 3H estimates (S5 Fig). On data simulated without MH, all three models (1H, 2H, 3H) yielded branch length estimates that were nearly identical, and a slight underestimate of the true values due to a bias in estimating the length of very short branches. However, on data simulated with MH, 1H models overestimate branch lengths compared to the 2H/3H models. A plausible explanation is that 1H models "expand" branch lengths slightly to compensate for the multi-hit events, while 2H and 3H models do not introduce bias in branch length estimates when there is no underlying multi-hit component because they can account for this situation by setting some parameters to 0.

## Simulations

**False positive rates.** We evaluated operating characteristics of the likelihood ratio tests (LRT) for MH model testing on parametrically simulated data. In the simplest case of a single-branch (two-sequence) null data generated under the 1H model, Type I error rates for 2H:1H and 3H:2H tests were on average below nominal. However, for very divergent sequences (e.g., >3 expected substitution per site), the test became somewhat anti-conservative, which is not surprising for such severely saturated data (S3 Fig). Individual branches that are this long are highly abnormal in real biological datasets. Expanded to multiple sequence alignments generated using parameter estimates from four biological datasets, simulations confirmed that all the tests employed appear to be somewhat conservative; this is by design because asymptotic distributions of LRT statistics when null hypotheses are on the boundaries of the parameter space are less conservative than the 1- or 2-degrees of freedom $\chi_d^2$ distributions we use here [34].

**Power.** The tests are generally well-powered, especially if the effect sizes (magnitudes of MH rates) are sufficiently large (Table 5). The power to detect two-hit substitution (2H:1H) is especially high (>90%) across all simulations. The test which attempts to identify non-zero triple-hit synonymous island rates (3HSI:2H) is the least powerful, because its signal is derived from a tiny fraction of all substitutions (substitutions between synonymous islands), i.e. the effective sample size is smaller that for the other tests.

**False positives due to alignment errors.** Whelan and Goldman [12] suggested that non-zero estimates of triple-hit rates could be at least partially attributed to alignment errors. It is impossible, with a few rare exceptions, to declare that any particular alignment of biological sequences is correct. Hence, in order to estimate what, if any, effect potential multiple sequence alignment errors might have on our inference, we simulated null data (1H model)

**Table 5. Power to detect MH rates.**

| Test | All | | Large effect | |
|------|-----|-----|-----|-----|
| | N | Power | N | Power |
| 2H:1H | 1956 | 94% | 967 | 99%($\delta > 0.5$) |
| 3H+:2H | 1940 | 64% | 1056 | 83%($\psi > 0.5$) |
| 3HSI:2H | 447 | 33% | 114 | 51%($\psi_s > 5.0$) |
| 3H+:3HSI | 1940 | 66% | 1056 | 86%($\psi > 0.5$) |

The fractions of simulated datasets that had $p < 0.05$ for the corresponding test. $N$ = number of simulations in each category, and the explicit definition effect size is shown in parentheses.

with varying indel rates with `Indelible` [28], inferred multiple sequence alignments `MAFFT` [35] in a codon-aware fashion, inferred trees using neighor-joining, and performed our hierarchical model fit. This procedure induces multiple levels of model misspecification, and errors: `Indelible` uses a different model (GY94 M3) to simulate sequences, there is alignment error, and there is phylogeny inference error. Sufficiently high indel rates coupled with other inference errors can indeed bias our tests to become anti-conservative, although these levels are higher than what we see (based on per-sequence "gap"/character) ratios for our biological alignments (S4 Fig). Empirical alignments have gap content that is consistent with alignments simulated with $0.01 - 0.015$ indel rates, for which test performance is nominal. The Selectome dataset [24] can also be retrieved masked for alignments with unreliability at certain sites. We compared the masked and unmasked alignments and found similar results: 76% of alignments in both cases favor 2H over 1H; 77% of unmasked alignments and 74% of masked alignments choose 3H over 1H (Table 4). While there is certainly some false signal due to mis-alignment, it is unlikely to be the dominant factor here. Nonetheless, care must be taken not to over-interpret MH findings when the alignments are uncertain.

## Discussion

Nearly three in five empirical alignments considered here provide strong statistical support that at least some of the substitutions are not well modeled by standard codon models. More than one in five prefer to have direct three-hit substitutions accounted for explicitly in the models. Substitutions involving serine codons, which are unique among the amino-acids in that they comprise two islands which are two or three nucleotide changes from each other, are prominent in driving statistical signal for these preferences, especially if they occur along short branches. Many other amino-acid pairs are also involved in such exchanges, indicating that not all of the statistical signal is due to serine codons, although in a typical alignment only a small fraction of sites (about 1%) prefer multiple hit models strongly.

Many previous studies have provided evidence that evolutionary models with multiple hits provide better fit to the data, but the scale of this phenomenon in the comparative evolutionary context has not been fully appreciated, although the interest in model development in this area is being rekindled. Our results also show that the inclusion of multiple-hit model parameters changes $\omega$ estimates, and with them—potentially alter inferences of positive selection, which was demonstrated for branch-site tests, [20], and for data simulated with multiple hits but analyzed with standard models [14]. Additionally, traditional models may slightly overestimate branch lengths for data where multiple-hit models provide better fits.

How much of this apparent support for multiple-hits comes from biological reality, and how much from statistical artifacts, or other unmodeled evolutionary processes—the so-called

phenomenological load [36]? Our simulation studies provide compelling evidence that the tests we use here are statistically well-behaved and possess good power, i.e. our positive findings are unlikely to be primarily a result of statistical misclassification. Other confounders, especially alignment error, have the potential to mislead the tests, but only at levels that appear higher than what is likely present in most biological alignments. In addition, there are some datasets (e.g., HIV reverse transcriptase), where alignment is not in question due to low biological insertion/deletion rates or structural information, and these data still support non-zero multiple-hit rates as well.

There is an abundance of data and examples of doublet substitutions in literature, and mechanistic explanations for them, e.g., due to polymerase zeta [7] exist. There are several papers arguing that the numbers of apparent triple-hits occurring in sequences is greater than what we would expect solely from random mutation [37–39], however the mechanism (if it exists) by which they might occur is speculative. As one option, Sakofsky et al. [40] have suggested that DNA repair mechanisms could help explain multi-nucleotide mutations.

Our analyses indicate that much, but not all, of the support for non-zero triple-hit rates derives from serine codon island jumping, particularly in cases when these jumps must occur over a short branch in the tree. Comparative species data might lack the requisite resolution to discriminate between instant multiple base changes and a rapid succession of single nucleotide changes spurred on by selection; the literature is split on which mechanism is primal [5, 6]. Such a common phenomenon is worth further investigation, in our opinion.

Our evolutionary models are broadly comparable to several others that have been published in this domain, some of which have more parametric complexity [14], or consider effects of substitutions spanning codon boundaries [12]. Our novel contributions are direct tests for the importance of synonymous island jumping, and a simple evidence ratio approach to identify and categorize specific sites that benefit from non-zero multiple hit rates. These models are easy to fit computationally, with roughly the same cost as would be required for an $\omega-$based positive selection analysis, and we provide an accessible implementation for researchers to use them. Further modeling extensions, e.g. the inclusion of synonymous rate variation, branch-site effects, etc., can be easily incorporated.

## Supporting information

**S1 Table. Estimated $\omega$ rate distributions for benchmark datasets for different models on the benchmark datasets.** $E[\omega]$: the mean $\omega$ value for the 1H model. $\frac{E[\omega]:2H}{E[\omega]:1H}$: the ratio of mean $\omega$ estimates from 2H and 1H models. $\frac{\delta:3H+}{\delta:2H}$: the ratio of $\delta$ estimates from 3H+ and 2H models. The datasets are sorted by increasing values of the $\frac{E[\omega]:2H}{E[\omega]:1H}$ column. Genes where there was significant evidence (LRT $p < 0.05$) of non-zero 2H rates are bolded, and those where there is evidence of non-zero 3H rates is underlined.
(PDF)

**S1 Fig. The effect of model choice on rate estimates.** Point estimates of global rate parameters under different models for each of the empirical datasets.
(TIF)

**S2 Fig. The fraction of sites with strong MH model preference.** Histograms are over alignments where there was significant ($p < 0.01$) support for the corresponding model: 20, 338 for 2H:1H (gray) and 7664 for 3H:2H (red).
(TIF)

**S3 Fig. False positive rates for LRTs on simulated data.** Results are shown for two sequence analyses (left) and multiple sequence analyses (right). For the two sequence simulations, we stratified the simulations by the length of the branch, *T*, (the range is labeled in the figure) measured in expected substitutions per site. The dotted line shows the nominal expectation (rejection rate = nominal p-value).
(TIF)

**S4 Fig. Indel rate verse TH rate.** Alignments with indel were simulated using INDELible across using the Dropsophila *a*dh tree and alignment length using GY94 M3 model with site-to-site $\omega$ variation. LRT p-values and rejection rates (FPR, at $p \leq 0.05$) are shown for different tests in the top row. The bottom row shows estimated $\delta$ and $\psi$ rates as a function of simulated indel rates, as well as the number of sites inferred to have high evidence ratios (ER) for 2H or 3H modes. The plot on the bottom right shows the average fraction of a sequence that in an alignment that is comprised of gaps is shown for simulated data, and empirical collections.
(TIF)

**S5 Fig. Branch-length estimate behavior under different models.** Simulated Data (1H): null simulations (1H model). Simulated Data (MH): power simulations. Selectome: empirical data. Red lines are drawn with least squares linear regression whose estimates slopes and intercepts as well as proportions of variance ($R^2$) explained are added to each plot.
(TIF)

## Author Contributions

**Conceptualization:** Alexander G. Lucaci, Sadie R. Wisotsky, Sergei L. Kosakovsky Pond.

**Data curation:** Alexander G. Lucaci, Sadie R. Wisotsky, Sergei L. Kosakovsky Pond.

**Formal analysis:** Alexander G. Lucaci, Sadie R. Wisotsky, Sergei L. Kosakovsky Pond.

**Funding acquisition:** Sergei L. Kosakovsky Pond.

**Investigation:** Alexander G. Lucaci, Sadie R. Wisotsky, Sergei L. Kosakovsky Pond.

**Methodology:** Alexander G. Lucaci, Sadie R. Wisotsky, Sergei L. Kosakovsky Pond.

**Project administration:** Alexander G. Lucaci, Sadie R. Wisotsky, Sergei L. Kosakovsky Pond.

**Resources:** Alexander G. Lucaci, Sadie R. Wisotsky, Sergei L. Kosakovsky Pond.

**Software:** Alexander G. Lucaci, Sadie R. Wisotsky, Sergei L. Kosakovsky Pond.

**Supervision:** Alexander G. Lucaci, Sadie R. Wisotsky, Sergei L. Kosakovsky Pond.

**Validation:** Alexander G. Lucaci, Sadie R. Wisotsky, Sergei L. Kosakovsky Pond.

**Visualization:** Alexander G. Lucaci, Sadie R. Wisotsky, Stephen D. Shank, Steven Weaver, Sergei L. Kosakovsky Pond.

**Writing – original draft:** Alexander G. Lucaci, Sadie R. Wisotsky, Sergei L. Kosakovsky Pond.

**Writing – review & editing:** Alexander G. Lucaci, Sadie R. Wisotsky, Sergei L. Kosakovsky Pond.

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
