## [Decision Letter · Decision Letter 0]

4 Dec 2020

PONE-D-20-31568

Extra base hits: widespread empirical support for instantaneous multiple-nucleotide changes.

PLOS ONE

Dear Dr. Lucaci,

Thank you for submitting your manuscript to PLOS ONE. After careful consideration, we feel that it has merit but does not fully meet PLOS ONE’s publication criteria as it currently stands. Therefore, we invite you to submit a revised version of the manuscript that addresses the points raised during the review process.

As you will see, both reviewers found the writing insufficiently clear and precise. Please take this comment seriously, as it is one of the key criteria of PLOS One, and it is impossible to evaluate well the science if the writing is unclear. In your revision, the key point is thus to improve the writing, but please address all comments by the reviewers.

We look forward to receiving your revised manuscript.

Kind regards,

Marc Robinson-Rechavi

Academic Editor

PLOS ONE

Journal Requirements:

Reviewers' comments:

Reviewer's Responses to Questions

**Comments to the Author**

1. Is the manuscript technically sound, and do the data support the conclusions?

Reviewer #1: Yes

Reviewer #2: Yes

2. Has the statistical analysis been performed appropriately and rigorously? 

Reviewer #1: Yes

Reviewer #2: Yes

3. Have the authors made all data underlying the findings in their manuscript fully available?

Reviewer #1: Yes

Reviewer #2: Yes

4. Is the manuscript presented in an intelligible fashion and written in standard English?

Reviewer #1: No

Reviewer #2: No

5. Review Comments to the Author

Reviewer #1: Lucaci et al present an empirical analysis to assess how widespread single vs multi-nucleotide hits are. On the whole, I find this research conceptually interesting, timely, and reasonable for publication. However, I found this paper quite difficult to review. Far too much of the paper is not written in complete sentences (the ideal number of incomplete sentences is 0), references appear to use the wrong LaTeX command (e.g. [5] should not be used as a sentence subject..), acronyms are not clearly defined when introduced (and from my reading seemed to shift in their definition? or introduced and then called something else later?), and the presentation of the model itself is very confusing. The proposed model itself is NOT that complex, so it is troubling that the presentation is not straightforward to follow.

Further, I have some difficulty clearly interpreting some of figures. For example, the monotone splines in Figure 2 are very misleading especially given that it seems there are wildly different sample sizes across the X-axis. I do not see a clear scientific interpretation of a monotone spline in this context. Fig S3 is ambiguously labeled, the double axes on Fig 4 (with what seem to be semi-outlandish ranges for the left-hand axis?) are virtually impossible to interpret reliably.

Based on my reading, the current manuscript version suffers from an overall lack-of-editing. It was not submitted in an "polished" form. I will be glad to review another version after the text has been significantly cleaned, properly organized, and figures have been designed to yield clear interpretations. Again, I believe the science here is solid, but it is not currently presented clearly at all.

Reviewer #2: Lucaci et al present codon substitution models that allow multiple nucleotide substitution in a codon. They show that such models provide better fit in LRTs. The authors show the results in a sensible flow, but I feel that the picture is quite incomplete.

According to the study, it seems that the majority of datasets and positions across an alignment do have preference for more complex models. It does make sense that more parameter-free models fit the data better, but I was not convinced that they produce better trees. Previous studies show that simpler models lead to more accurate branch-length estimates. I wonder whether these results are biased by the hypothesis testing method. I would suggest to compare the resulting trees of the simulation studies to the starting trees and look at the distances, e.g., branch-length distance. It could be that 3H models are better fitted but generate overestimated branch lengths whereas 1H models are more conservative and produce more accurate trees.

Other than that, I think that the clarity of the text, especially in the Introduction and Methods, could be improved. I had a hard time understanding the details before I reached the Results section. For example, models notations in the first paragraph of the Methods section are only defined in the Results.

6. PLOS authors have the option to publish the peer review history of their article (what does this mean?). If published, this will include your full peer review and any attached files.

Reviewer #1: No

Reviewer #2: No

---

## [Author Response · Author response to Decision Letter 0]

15 Jan 2021

We have responded to reviewers comments and have attached our revised manuscript and letter.

---

## [Decision Letter · Decision Letter 1]

10 Feb 2021

PONE-D-20-31568R1

Extra base hits: widespread empirical support for instantaneous multiple-nucleotide changes.

PLOS ONE

Dear Dr. Lucaci,

Thank you for submitting your manuscript to PLOS ONE. I am sorry for the delay. Reviewer 1 has clarified that they no longer wished to review this manuscript, so I took time to evaluate it myself rather than lose even more time finding another reviewer. After careful consideration, we feel that it has merit but does not fully meet PLOS ONE’s publication criteria as it currently stands. Therefore, we invite you to submit a revised version of the manuscript that addresses the points raised during the review process.

I agree with reviewer 2 that the work is ready for publication in PLOS One, but that the writing should be carefully checked. In addition to the errors noted by the reviewer, I have noticed the following:

"signal due to misaligned" misses a word, probably "misaligned sites".

"for one 384 branch-site tests, [20]" if there is one test it shouldn't have an 's'.

We look forward to receiving your revised manuscript.

Kind regards,

Marc Robinson-Rechavi

Academic Editor

PLOS ONE

Reviewers' comments:

Reviewer's Responses to Questions

**Comments to the Author**

1. If the authors have adequately addressed your comments raised in a previous round of review and you feel that this manuscript is now acceptable for publication, you may indicate that here to bypass the “Comments to the Author” section, enter your conflict of interest statement in the “Confidential to Editor” section, and submit your "Accept" recommendation.

Reviewer #2: All comments have been addressed

2. Is the manuscript technically sound, and do the data support the conclusions?

Reviewer #2: Yes

3. Has the statistical analysis been performed appropriately and rigorously? 

Reviewer #2: Yes

4. Have the authors made all data underlying the findings in their manuscript fully available?

Reviewer #2: Yes

5. Is the manuscript presented in an intelligible fashion and written in standard English?

Reviewer #2: Yes

6. Review Comments to the Author

Reviewer #2: The authors have addressed my concerns. The flow of the manuscript was much improved and it is now much easier to follow and understand.

There are several English mistakes in some parts of the text - I suggest that the authors would thoroughly review the writing. Examples: "have have" in line 20; “traction” should be “attraction” in line 45; "these models *have* been unable" in line 44).

7. PLOS authors have the option to publish the peer review history of their article (what does this mean?). If published, this will include your full peer review and any attached files.

Reviewer #2: No

---

## [Author Response · Author response to Decision Letter 1]

19 Feb 2021

We have made all changes as requested in this minor revision.

---

## [Editor Report · Decision Letter 2]

25 Feb 2021

Extra base hits: widespread empirical support for instantaneous multiple-nucleotide changes.

PONE-D-20-31568R2

Dear Dr. Lucaci,

We’re pleased to inform you that your manuscript has been judged scientifically suitable for publication and will be formally accepted for publication once it meets all outstanding technical requirements.

Kind regards,

Marc Robinson-Rechavi

Academic Editor

PLOS ONE
---

## [Editor Report · Acceptance letter]

3 Mar 2021

PONE-D-20-31568R2 

Extra base hits: widespread empirical support for instantaneous multiple-nucleotide changes. 

Dear Dr. Lucaci:

I'm pleased to inform you that your manuscript has been deemed suitable for publication in PLOS ONE. Congratulations! Your manuscript is now with our production department. 

Kind regards, 

on behalf of

Prof. Marc Robinson-Rechavi 

Academic Editor

PLOS ONE